# Comparison of Activity of Commercial Protective Cultures and Thermophilic Lactic Acid Bacteria against *Listeria monocytogenes:* A New Perspective to Improve the Safety of Sardinian PDO Cheeses

**DOI:** 10.3390/foods12061182

**Published:** 2023-03-10

**Authors:** Maria Pina Meloni, Francesca Piras, Giuliana Siddi, Daniela Cabras, Eleonora Comassi, Roberta Lai, Olivia McAuliffe, Enrico Pietro Luigi De Santis, Christian Scarano

**Affiliations:** 1Department of Veterinary Medicine, University of Sassari, Via Vienna, 2, 07100 Sassari, Italy; 2Teagasc Food Research Centre, Moorepark, Fermoy, P61 C996 Co. Cork, Ireland

**Keywords:** *L. monocytogenes*, bioprotective cultures, cheese, lactic acid bacteria, agar well diffusion assay

## Abstract

*Listeria monocytogenes* contamination that occurs during and post-processing of dairy products is a serious concern for consumers, and bioprotective cultures can be applied to control the growth of the pathogen in sheep milk cheeses. However, to respect specifications provided for protected designation of origin (PDO) cheeses, only autochthonous microorganisms can be used as bioprotective cultures in these products. This in vitro study aimed to evaluate thermophilic lactic acid bacteria (LAB) isolated from sheep milk as bio-preservative agents to control *L. monocytogenes* growth in PDO cheese. Results were compared with those obtained with a commercial protective culture (cPC) composed of a *Lactiplantibacillus plantarum* bacteriocin producer designed to inhibit *L. monocytogenes* growth in cheese. The in vitro antilisterial activities of n.74 autochthonous LAB and a cPC were tested against 51 *L. monocytogenes* strains using an agar well diffusion assay. In addition, 16S rRNA sequencing of LAB isolates with antilisterial activity was conducted and strains of *Lactobacillus helveticus*, *Lactobacillus delbrueckii* subsp. *indicus*, *Lactobacillus delbrueckii* subsp. *sunkii*, *Lactobacillus delbrueckii* subsp. *lactis* and *Enterococcus faecalis* were identified. In this study, 33.6% (74/220) bacterial strains isolated from milk had characteristics compatible with thermophilic LAB, of which 17.6% (13/74) had in vitro antilisterial activity. These results demonstrate that raw sheep milk can be considered an important source of autochthonous thermophilic LAB that can be employed as protective cultures during the manufacturing of Sardinian PDO cheeses to improve their food safety. The use of bioprotective cultures should be seen as an additional procedure useful to improve cheese safety along with the correct application of good hygienic practices during manufacturing and the post-processing stages.

## 1. Introduction

*Listeria monocytogenes* is a Gram-positive, psychrotrophic, foodborne pathogen responsible for listeriosis. In the European Union (EU), an estimated 1600 people contract listeriosis each year, with approximately 260 fatalities [1]. This makes listeriosis one of the most serious foodborne diseases and the fifth most widespread zoonosis in the EU [2]. The severe infection is most likely to impact pregnant women and their newborns, adults aged 65 or older, and people with weakened immune systems [1]. In 2020, 1876 invasive human cases of *L. monocytogenes* were confirmed. The case fatality rate was as high as 13.0%, but decreased compared with 2019 and 2018. The 26% decrease in the incidence of foodborne pathogen infections detected in 2020 could be due to health interventions to prevent the transmission of SARS-CoV-2. As a result of travel restrictions, infections associated with international travel decreased, and the closure of restaurants may also have helped reduce the incidence of these diseases. However, this decrease may also be due to fewer samples being tested due to the pandemic [3].

*L. monocytogenes* is ubiquitous and able survive and grow in a range of natural environments [4] in temperatures ranging between 0 and 45 °C, pH between 4.4 and 9.4, and a minimum water activity (aw) value of 0.92. Foods with a long shelf-life stored under refrigeration temperatures and ready-to-eat (RTE) foods, i.e., food intended for direct consumption without the need for cooking or other processing steps that would eliminate the pathogen, are most often associated with listeriosis [5]. Ready-to-eat foods such as cheeses may be contaminated by *L. monocytogenes* during and after production in the phases of ripening, storage, or packaging [6], and the pathogen can reach levels dangerous for human health during distribution and storage. *L. monocytogenes* contamination is commonly reported in dairy products, including soft and semi-soft cheeses, and raw milk and raw milk products are often incriminated as causes of listeriosis outbreaks [7,8].

For these reasons, the dairy industry and public health authorities are paying particular attention to dairy products as a major source of listeriosis outbreaks [9] and regulations have been put in place to control the growth of *L. monocytogenes* in foods [10]. Commission Regulation (E.C.) No 2073/2005 on microbiological criteria for foodstuffs establishes food safety criteria for *L. monocytogenes.* In foods that can support the growth of the organism, it must normally be absent in 25 g. In the case of ready-to-eat foods unable to support the growth of *L. monocytogenes*, other than those intended for infants and special medical purposes, this limit is raised to 100 CFU/g during the expected or stated length shelf life [11].

The sheep and goat economic sector accounts for 1.2% of total Italian agricultural production, with over 6.7 million animals [12,13]. With 75.8 thousand tons produced annually, Italy is the primary producer of sheep milk cheese in the European Union [14]. The sheep dairy sector is particularly relevant in the economic performance of certain regions such as Sardinia, where 56% of the national sheep and goat population is located [15]. Ten percent of European sheep milk and 60,000 tons of sheep milk cheese are produced in Sardinia and about 30,000 tons of these are protected designation of origin (PDO) cheeses (mostly Pecorino Romano PDO). Sardinia could therefore be considered at the same level of countries such as Spain and precedes others such as France and Portugal in terms of production [16]. The European Union has awarded the PDO label to over 180 cheeses and, of the 50 Italian PDO cheeses, 3 are produced in Sardinia: “Pecorino Romano”, “Fiore Sardo”, and “Pecorino Sardo” [17]. In order to respect the PDO specifications, these cheeses are produced using heat-treated sheep milk (for the production of *Pecorino Romano* and *Pecorino Sardo*) or raw sheep milk (for the production of *Fiore Sardo*) [18]. The milk must originate exclusively from the production area and it may be inoculated with natural and autochthonous starter cultures [19,20].

Over time, as a consequence of food safety requirements, measures have been designed with the aim of protecting consumer health to prevent or reduce the health hazards derived from food, in particular for ready-to-eat products. Foodborne pathogens, such as *L. monocytogenes*, can contaminate foods, including cheeses, thus posing a risk to consumers; therefore, strict guidelines have been established to control foodborne pathogens in cheese [21]. For this reason, there is an increasing need for innovative strategies to improve food safety while respecting tradition and production technologies. In recent years, bioprotective microorganisms have been proposed as an alternative antimicrobial strategy to improve food safety [22]. The use of lactic acid bacteria to produce antimicrobials such as bacteriocins, hydrogen peroxide, ethanol and other organic acids to control foodborne pathogens is an increasingly widespread strategy in food industries [23]. Lactic acid bacteria (LAB) that exhibit inhibition of undesired microorganisms are defined as “Protective Cultures”; they can improve the microbiological stability of foods and reduce the risk of growth and persistence of food-borne pathogens and food spoilage organisms [24].

LABs are also physiologically present in the gastro-intestinal tract of humans and animals and are an essential part of the gut microbiome [25,26,27]. These characteristics make certain LAB strains excellent candidates for use in ready-to-eat products such as cheese, and numerous studies demonstrated their effectiveness against *L. monocytogenes* growth [23,28,29,30]. 

Commercially available protective cultures, composed of selected LABs, are considered safe and easy to use in food, but they also have some limitations. In fact, PDO regulations forbid the addition of substances or microorganisms that are not derived from the product itself [31]. Moreover, previous experiments using a mesophilic commercial protective culture (such as *L. plantarum*) found that the high temperatures applied during some production processing steps (45–47 °C) were more appropriate for thermophilic LABs than for mesophilic cultures [32].

Based on these data and with the aim of introducing protective cultures into the production processes of PDO cheeses, the present study aimed to isolate, identify, and select autochthonous thermophilic LABs isolated from Sardinian raw sheep milk that could potentially be used as protective cultures against *L. monocytogenes* in Pecorino Sardo PDO cheese. This preliminary study also compared the antimicrobial activity of one commercial protective culture to these autochthonous thermophilic LABs against *L. monocytogenes* and evaluated their adaptability to cheese production.

## 2. Materials and Methods

### 2.1. Experimental Design

An experiment was designed to compare a commercial protective culture (cPC) and autochthonous thermophilic LAB with anti-listerial activity, with the following four steps:Step one: Isolation, morphological and biochemical characterisation of LABs. Bacteria with morphological and biochemical characteristics typical of LAB and able to grow at 45 °C were isolated from Sardinian raw sheep milk;Step two: In vitro assessment of cPC and LABs antilisterial activity. The antilisterial activities of cPC and LAB cultures were tested and compared using an agar well diffusion assay (AWDA) method.Step three: Investigation of the mode of action of cPC and LABs. AWDA using cPC and LAB cell-free supernatants was conducted.Step four: Evaluation of antilisterial activity and viable count enumeration of cPC and LABs after growth in “*scotta*” (the residual whey from ricotta cheese production). cPC and LAB antilisterial activities were tested using *scotta* as growth media to investigate their adaptability to the manufacturing technology of the cheese.

### 2.2. Step One: Isolation, Morphological and Biochemical Characterisation of LABs

To isolate potential antilisterial LAB from Sardinian raw sheep milk, four samples of milk, each containing a milk pool from the milk transport tanks, were collected from a local cheese making plant. These raw milk pool samples were collected from several primary production holdings located in different areas of Sardinia and pooled in milk transport tanks, from which the samples were taken. All the milk samples were collected in sterilised bottles, transported to the laboratory at controlled temperature (4 ± 1 °C) and used within 2 h of sampling.

Four pools of raw milk were serially diluted from 10^−1^ to 10^−4^ by transferring 1 mL (1:10 dilution) to 9 mL sterilised 0.85% (w v^−1^) NaCl [33]. Serial dilutions were spread into Petri dishes containing Man Rogosa and Sharpe (MRS, Biolife, Milano, Italy) agar (pH 5.5, adjusted with 0.5 N HCl) for isolation of lactobacilli, M17 (Microbiol, Cagliari, Italy) agar with lactose 0.5% (Farmitalia Carlo Erba S.p.A., Milano, Italy) for isolation of lactococci (45 °C incubation, aerobic and anaerobic conditions) and Elliker (Sigma-Aldrich, St. Louis, Missouri, USA) agar with 0.5% beef extract (Biolife, Milano, Italy) for cultivation and enumeration of *Streptococcus thermophilus* [34,35,36]. All experiment stages were performed in triplicate. Statistical analysis on the set of data obtained was performed by calculating the mean and standard deviation of the numbers of colonies obtained to measure the differences of each observation from the mean; in addition, the data obtained were compared using statistical analysis of variance (ANOVA, ANalysis Of VAriance). Morphologically distinct colonies were submitted to Gram stain evaluation and catalase and oxidase tests. The Gram-positive and catalase-negative isolates were selected as presumptive LABs [37], later confirmed using 16S rRNA gene sequencing. The isolates were stored at −20 °C and −80 °C in MRS, L-M17, or B-Elliker broth containing 10% glycerol for further analysis and molecular identification.

### 2.3. Step Two: In Vitro Assessment of cPC and LABs Antilisterial Activity

The antilisterial activities of cPC and LAB isolates from step 1 were evaluated against 51 *L. monocytogenes* strains. Of these 51 strains, 47/51 (92.1%) were previously isolated from Sardinian cheese-making plants (both from dairy products and the environment) [38], while the others were reference strains (ATCC19111, NCTC1088, ATCC764, ATCC19115).

The cPC “Lyofast LPAL” (*Lactiplantibacillus plantarum*) manufactured by Sacco System (Cadorago, Italy) and 74 autochthonous thermophilic presumptive LABs were initially screened via agar well diffusion assay (AWDA). cPC and autochthonous LABs were inoculated in their specific growth media and incubated overnight at the previously described conditions.

For the AWDA, an overnight culture of each of the *L. monocytogenes* strains was inoculated into Brain Heart Infusion agar (BHA, Biolife, Milano, Italy) to obtain a final concentration of 10^5^ CFU/mL and poured into Petri dishes [39]. Wells of 8 mm diameter were in the agar plates, and 50–100 µL of cPC or LAB cultures was added to each well [40,41]. The plates were then incubated at 37 °C for 24 h [42]. The presence of an inhibition zone around the wells was considered as a positive for antilisterial activity.

### 2.4. Step Three: Investigation of the Mode of Action of cPC and LABs

In order to distinguish between antibacterial activity related to bacteriocins or bacteriocin-like substances, an AWDA protocol was conducted. Overnight cultures of cPC and LABs identified as effective against *L. monocytogenes* in the previous experiment were centrifuged at 14,000× *g* for 5 min [43]. The cell-free supernatants (CFSs) were adjusted to pH 6.5 with 1 M NaOH in order to rule out acid inhibition [44,45] and subsequently treated with 1 mg mL^−1^ of catalase at 25 °C for 30 min to eliminate the possible inhibitory action of hydrogen peroxide. Plates containing semi-solid BHA previously inoculated with 10^5^ CFU/mL of *L. monocyotogenes* [39] were prepared and AWDA was performed, inoculating wells with CFS filtered using a 0.22 µm filter (Millex SLGP033RS, Millipore, Bedford, MA, USA) [46]. Plates were examined after incubation at 37 °C for 24 h [47] and inhibition zone diameters were measured.

To confirm the protein nature of the antilisterial compound produced by cPC as described by the producer, its CFS was treated with the enzyme proteinase K. Proteinase K is a highly active protease, stable over a wide range of pHs and capable of withstanding high temperatures. To evaluate the proteolytic susceptibility of bacteriocins produced by cPC (*L. plantarum*), its CFS was treated at 37 °C for 2 h with proteinase K at a final concentration of 1 mg/mL. After adjusting the pH to 6.5, the samples were filtered through a 0.22 μm sterile filter (Millipore) and the antimicrobial activity was determined by the AWDA method as previously described [46].

The best-performing LAB strains, meaning those that produced the largest inhibition zone, were used for other experiments described below.

### 2.5. Step Four: Evaluation of Antilisterial Activity and Viable Count Enumeration of cPC and LABs after Growth in “Scotta”

The seven autochthonous thermophilic presumptive LABs that exhibited the best in vitro activity against *L. monocytogenes* were selected and tested, along with the cPC, for their ability to grow in *scotta*. Following the same procedure used to prepare the *scotta-innesto* (the typical starter culture) [48], a modified *scotta-innesto* (MSI) was prepared: cPC and LABs were suspended in *scotta* and incubated overnight at 37 °C for cPC and 45 °C for LABs. For cPC and LAB enumeration, 1 mL of the appropriate dilution was transferred into Petri dishes and 10–12 mL of their selective growth agar media was added. Finally, plates were incubated in anaerobic conditions for 72 h at 30 °C and 45 °C, respectively [49,50]. Data obtained were statistically analysed by calculating the mean and standard deviation and compared using statistical analysis of variance (ANOVA, ANalysis Of VAriance). 

AWDA was performed as previously described with minor modifications. Briefly, BHA spiked with *L. monocytogenes* at a final concentration of 10^5^ CFU/mL was poured into Petri dishes. Then, wells were cut in the agar and filled with the overnight cultures of *scotta* inoculated with cPC and LABs. These plates were incubated for 24 h at 37 °C and inhibition zones were measured [51]. All experiments were performed in triplicate.

### 2.6. Molecular Identification of Autochthonous Thermophilic Presumptive Lactic Acid Bacteria

Seven autochthonous thermophilic presumptive LABs that demonstrated antilisterial activity after AWDA were sent to B.M.R. Genomics (Padova, Italy) for 16S rRNA gene sequencing to identify isolates at the species level according to the protocol of Clarridge [52]. B.M.R. Genomics performed 16S gene amplification and sequencing, after which a search for the homology of the sequences was made using the BLAST algorithm [53,54] available from the NCBI server URL: http://blast.ncbi.nlm.nih.gov/Blast.cgi (accessed on 1 December 2022).

## 3. Results

### 3.1. Step One: Isolation, Morphological and Biochemical Characterisation of LABs

Autochthonous thermophilic bacterial isolates were obtained from Sardinian raw sheep milk following cultivation on MRS pH 5.5, L-M17, and B-Elliker media. Table 1 and Table 2 list the viable counts obtained.

Viable colony counts in aerobic conditions in MRS pH 5.5 were significantly lower (*p* < 0.01) than in L-M17 and B-Elliker, while no significant difference was observed between L-M17 and B-Elliker (*p* > 0.05).

For bacteria grown in anaerobiosis, the situation was similar to that of bacteria grown in aerobic conditions. The bacterial count in MRS pH 5.5 was significantly lower (*p* < 0.01) than in L-M17 and B-Elliker.

Comparing the two growth conditions used (aerobic and anaerobic), the bacterial counts in L-M17 and B-Elliker were significantly higher (*p* < 0.01) in the samples grown in anaerobiosis. No significant differences were found for samples grown in MRS pH 5.5.

From sheep milk, we isolated 220 strains, and 102/220 strains demonstrated a catalase-negative reaction. The Gram stain demonstrated that 28/102 were Gram-negative and 74/102 were Gram-positive bacteria; 92/102 were classified as oxidase-negative. In addition, 49/102 isolates were rod-shaped and the remaining 53/102 were coccus-shaped. The resulting 74/220 (33.6%) Gram-positive and catalase-negative strains were evaluated for their inhibitory activity against *L. monocytogenes* using the agar well diffusion assay method.

### 3.2. Step Two: In Vitro Assessment of cPC and LAB Antilisterial Activity

After overnight growth in their specific culture media, the cPC demonstrated in vitro antilisterial activity against all 51 *L. monocytogenes* strains tested and produced an inhibition halo with a diameter > 2.1 cm. We found that 13/74 (17.6%) autochthonous presumptive LABs demonstrated antimicrobial activity against *L. monocytogenes* and, of these, 7/13 (53.8%) produced an inhibition halo with a diameter ranging from 1.1 cm to 2.0 cm, while the remaining 6 LABs had an inhibition halo with a diameter between 0.1 and 1 cm. Inhibition halos produced by cPC and LABs were significantly different (*p* < 0.01). Therefore, the seven LABs that demonstrated the highest efficacy against *L. monocytogenes* were used in the next phases.

### 3.3. Step Three: Investigation of cPC and LAB Modes of Action

Using CFS, cPC produced an inhibition halo with a diameter > 2.1 cm in AWDA, while the same experiment demonstrated negative results for all LAB strains. With regard to the evaluation of proteinase K’s effect on the cPC bactericidal activity, no anti-listeria activity after treatment with proteinase K was found. This result confirmed the manufacturer indications that report that the cPC *L. plantarum* strain is a bacteriocin producer.

These results confirmed that cPC antilisterial activity is related to bacteriocin production, as declared by the manufacturer. Among the bacterial strains isolated from raw sheep milk that were positive on the AWDA, 7/13 (53.8%) demonstrated the largest inhibition zone and therefore were chosen to perform further investigation.

### 3.4. Step Four: Evaluation of Antilisterial Activity and Viable Count Enumeration of cPC and LABs after Growth in “Scotta”

cPC and LAB viable counts were evaluated after overnight growth in *scotta* in order to investigate their adaptability to dairy matrices. The same samples grown in *scotta* were tested for their inhibitory activity against *L. monocytogenes* via AWDA and demonstrated significant differences (*p* < 0.01). Inhibition zone diameters were evaluated and compared with data obtained from the traditional AWDA, in which cPC and LABs grown in traditional media instead of *scotta* were used. The halo of inhibition was significantly higher (*p* < 0.01) when cPC grew in the traditional growth medium than in *scotta*. After growth in traditional media and in *scotta*, inhibition halos produced by LABs demonstrated no significant differences (*p* > 0.05). Results are listed in Table 3.

Statistical analysis of the results listed in Table 3 shows that the bacterial strains L1 MRS 1b and L2 MRS 9b grown in “*scotta*” demonstrated a more effective in vitro antilisterial activity compared to the same strains grown in traditional media, with a significantly larger inhibition halo diameter measured (*p* < 0.01). On the other hand, cPC demonstrated a significantly larger inhibition halo after growth in MRS pH 5.5 (*p* < 0.01). No significant differences were detected for the other strains (*p* > 0.05).

### 3.5. Molecular Identification of Autochthonous Thermophilic Presumptive Lactic Acid Bacteria

The results obtained by 16S rRNA gene sequencing confirm that the isolates were lactic acid bacteria. More specifically, the selected LABs with antilisterial activity in ADWA were identified as *Lactobacillus delbrueckii* subsp. *sunkii*, *Lactobacillus delbrueckii* subsp. *lactis*, *Lactobacillus delbrueckii* subsp. *indicus*, *Lactobacillus helveticus,* and *Enterococcus faecalis* (Table 4).

## 4. Discussion

Currently, much emphasis is placed on the safety of food production methods by producers, competent authorities, and consumers. Therefore, it is essential to search for strategies that guarantee food safety while, at the same time, maintaining the characteristics of specific products. To address this topic, this study focused on investigating the use of bioprotective cultures to inhibit the growth of *L. monocytogenes* in Sardinian PDO cheeses. Many studies have demonstrated the efficacy of commercially produced protective cultures and their potential for use in various food applications. The use of bioprotective cultures is a promising strategy for enhancing the safety of many kinds of food such as meat products, in which a significant reduction (*p* < 0.01) >1 Log_10_ CFU/g of *L. monocytogenes* has been demonstrated [55], as well as dairy products [56,57].

The main objective of this work was to select autochthonous LABs that could potentially be used as bioprotective cultures to improve the safety of traditional PDO cheeses, such as Pecorino Sardo DOP or other Sardinian DOP dairy products. LABs are called “autochthonous” to indicate their environmental and indigenous origin [58,59]. Raw milk is characterised by a dynamic and diverse microbial community. Bacteria in milk have been widely reported, but the majority of bioprotective strains have been isolated from human or bovine milk [60]. For this reason, the present study investigated Sardinian sheep milk, commonly used for cheese production in Sardinia, as a source of potential biopreservative lactic acid bacteria. During pasteurisation, which is an important step to eliminate milk-borne pathogens, the number of microorganisms, including LABs, is reduced 20-fold [60,61]. However, some of the bacteria that do not survive pasteurisation could have beneficial effects in dairy products, in particular with regard to the protection of consumer health, such as those stemming from their ability to produce antimicrobial compounds active against pathogens [62] and maintain and reinforce the human immune system [63]. In response to this problem, developments in scientific research have proposed some strategies, such as the applications of protective cultures [64,65]. The addition of autochthonous LABs during cheesemaking can represent a good compromise between the safeguarding of traditional products, the adoption of solutions that allow high levels of food safety, and the possibility to maintain traditional manufacturing technologies [66]. In agreement with the results of other investigations [67], this study demonstrated a prevalence of Gram-positive bacteria in raw sheep milk, almost equally divided between rod-shaped bacilli (50.7%) and cocci (49.3%). Of these bacteria, 33.6% were identified as presumptive LABs due to their morphological and biochemical features, and 13/74 (17.6%) demonstrated antilisterial activity in vitro. This result highlights that Sardinian sheep milk is a good source of bioprotective microorganisms that could substitute for commercial bioprotective cultures. The data reported in this work can be compared with other similar studies in which indigenous LABs were selected and used as bioprotective cultures in foodstuffs. Our results show that LABs isolated from Sardinian raw milk present antilisterial capability that is superior to both the results of Ortolani et al. (2010), who found that 14.9% LAB strains isolated from raw milk and soft cheese exhibited antilisterial activity [28], and Reuben et al. (2020), who observed that 6.5% of LABs isolates from goat and cow raw milk were inhibitory to *L. monocytogenes* [68]. Considering cheese as a source of bioprotective bacteria, further results on the percentages of indigenous LABs with antilisterial activities can be found in the literature [30,69,70]. 

Traditional food production, such as the production of PDO cheeses, is part of Sardinian regional culture and is very important in terms of the economic activity of the region. The production specification of Pecorino Sardo PDO provides that “the whole sheep milk is inoculated only with bacteria from the origin area” [71]. The use of native microorganisms with inhibitory action against *L. monocytogenes* allows improvements in food safety in compliance with PDO procedures, specifications, and guidelines. Use of autochthonous thermophilic LABs, e.g., lactobacilli, to inhibit foodborne pathogens potentially present in dairy products makes them suitable microorganisms for use in cheesemaking [72]. In this study, thermophilic autochthonous LABs were tested for two reasons: Firstly, thermophilic LABs are more tolerant of the temperatures encountered during cheese production than mesophilic bacteria. Secondly, it could be useful to introduce protective cultures during the cheesemaking process, as is commonly performed for starter cultures. In fact, bioprotective cultures are traditionally sprayed onto the cheese surface [64], but this step is time-consuming and requires specialised personnel. The use of microorganisms with bio-preservative potential are attractive to dairy industries, but there is the need for an easier mode of application for the food safety operators. To evaluate this, this study investigated the possibility of inoculating the protective microorganisms in the milk during the cheesemaking process rather than spraying post-production. *Scotta* was identified as a medium compatible with dairy processes. “*Scotta*” is a residual milk by-product of ricotta cheese production and is currently used in Sardinia to produce “*Scotta-innesto*”, a whey starter obtained by incubating starter bacteria in *scotta* at 42–45 °C for 18–20 h [48].

The use of *scotta* during sheep milk cheese production is a well-established procedure for preparing the *scotta-innesto* used for different kinds of Sardinian PDO cheese [73]. With this study, we evaluated whether *scotta* could be an appropriate medium to inoculate bioprotective cultures during PDO cheese production. In this work, a modified *scotta-innesto* (MSI) was created in which the starter culture was replaced with cPC and LABs. The results of AWDA show that in vitro antilisterial activity was maintained in the MSI. Therefore, there is potential in inoculating the milk with bioprotective cultures during the cheese-making process.

## 5. Conclusions

*L. monocytogenes* is one of the most important microorganisms of concern in the dairy sector. Therefore, identifying strategies to improve cheese safety is of paramount importance. Given the key role of the dairy sector in Sardinia, maintaining and improving the safety standards of Sardinian PDO cheeses becomes even more important, given the huge economic losses that could arise in the case of *L. monocytogenes* contamination. In addition, consumers are increasingly aware of food safety issues, but at the same time, require products that are free from chemical additives. Results of this study identify strategies that could meet the needs of dairy farms and the demands of the consumer and allow compliance with the criteria laid down in community legislation.

Autochthonous thermophilic LABs identified in the present study are excellent candidates for bio-protective cultures concerning Sardinian PDO cheeses. The results obtained in this study show that Sardinian sheep milk is an excellent source of lactic acid bacteria with antilisterial activity and that these LAB strains show the potential to improve food safety in compliance with PDO specifications. In accordance with previous studies, the MRS culture medium appeared the most appropriate for isolation and growth of bioprotective lactic acid bacteria [74,75]. Evaluation of in vitro cPC and LAB inhibitory activity against *L. monocytogenes* found the effectiveness of bioprotective lactic acid bacteria against this relevant pathogen for the dairy industry. Autochthonous *Lactobacillus* species could be considered for new potential bioprotective cultures; moreover, their excellent adaptability to dairy matrices such as *scotta* and their optimum growth temperature (45 °C) makes them particularly suitable to be used for inoculation during the cheesemaking processes. These preliminary results will be useful to carry out challenge tests conducted by inoculating the two types of bioprotective cultures (commercial and autochthonous) in milk during cheese production in order to assess whether the antilisterial activity demonstrated in vitro is confirmed in the cheese. The use of bioprotective cultures, in combination with good hygiene in food processing, may become a promising strategy to control pathogen contamination in cheeses and improve food safety. Despite the encouraging results obtained from this preliminary study, it is essential for producers and consumers to take further precautions to prevent the contamination of cheeses, especially during post-process handling.

## Figures and Tables

**Table 1 foods-12-01182-t001:** Viable colony counts (log_10_ CFU/mL, mean ± standard deviation) of thermophilic LABs from Sardinian raw sheep milk samples plated on MRS pH 5.5, L-M17, and B-Elliker in aerobic conditions at 45 °C.

Raw Sheep Milk Samples	Culture Media		
	MRS pH 5.5 ^4^	L—M17 ^5^	B—Elliker ^6^
L1 ^1^	2.1 ^2^ ± 0.3 ^3 a^	3.6 ± 0.0 ^b^	3.5 ± 0.0 ^b^
L2	3.0 ± 0.7 ^a^	3.5 ± 0.1 ^b^	3.7 ± 0.2 ^b^
L3	1.4 ± 0.8 ^a^	3.6 ± 0.2 ^b^	3.8 ± 0.2 ^b^
L4	2.7 ± 0.1 ^a^	3.2 ± 0.6 ^b^	3.7 ± 0.3 ^b^

^1^ Milk samples from different areas of Sardinia; ^2^ Log_10_ CFU/mL; ^3^ Values indicate the average viable count (Log_10_ CFU/mL) of triplicate experiments ± standard deviation; ^4^ Man Rogosa and Sharpe agar (pH 5.5, adjusted with 0.5 N HCl); ^5^ M17 agar with 0.5% lactose; ^6^ B-Elliker agar with 0.5% beef extract. Means in the same row with different letters were significantly different (*p* < 0.01).

**Table 2 foods-12-01182-t002:** Viable colony counts of thermophilic LABs from Sardinian raw sheep milk samples plated on MRS pH 5.5, L-M17, and B-Elliker in anaerobic conditions at 45 °C.

Raw Sheep Milk Samples	Culture Media		
	MRS pH 5.5 ^4^	L—M17 ^5^	B—Elliker ^6^
L1 ^2^	2.2 ^1^ ± 0.1 ^3 a^	4.6 ± 0.3 ^b^	4.3 ± 0.2 ^b^
L2	2.8 ± 0.4 ^a^	4.5 ± 0.2 ^b^	4.6 ± 0.2 ^b^
L3	2.3 ± 0.1 ^a^	3.9 ± 0.3 ^b^	4.3 ± 0.3 ^b^
L4	3.0 ± 0.2 ^a^	4.8 ± 0.2 ^b^	4.5 ± 0.2 ^b^

^1^ Milk samples from different areas of Sardinia; ^2^ Log_10_ CFU/mL; ^3^ values indicate the average viable count (Log_10_ CFU/mL) of triplicate experiments ± standard deviation; ^4^ Man Rogosa and Sharpe agar (pH 5.5, adjusted with 0.5 N HCl); ^5^ M17 agar with 0.5% lactose; ^6^ B-Elliker agar with 0.5% beef extract. Means in the same row with different letters were significantly different (*p* < 0.01).

**Table 3 foods-12-01182-t003:** The first column lists the analysed samples. The second column shows the viable count (log_10_ CFU/mL ± standard deviation) of cPC and the 7 best-performing LABs after overnight growth in *scotta*. The third column shows results of AWDA: antilisterial activities in vitro (diameter of the inhibition halo) of cPC and LAB grown in traditional media and in *scotta* are compared.

Samples	cPC Mean Viable Counts	Agar Well Diffusion Assay	
		After Growth in Traditional Media (cm)	After Growth in *Scotta* (cm)
Blank *scotta* (without inoculum)	0 ± 0.0 ^2^	-	-
cPC	8.81 ± 0.1	2.5 ± 0.3 ^a^	1.9 ± 0.3 ^b^
L4 M17 2A ^1^	7.60 ± 0.1	1.6 ± 0.1 ^a^	1.7 ± 0.2 ^a^
L1 MRS 1B	8.02 ± 0.2	1.2 ± 0.1 ^a^	1.1 ± 0.0 ^b^
L2 MRS 7B	7.66 ± 1.0	1.4 ± 0.2 ^a^	1.4 ± 0.2 ^a^
L2 MRS 8B	7.54 ± 0.6	1.4 ± 0.3 ^a^	1.6 ± 0.4 ^a^
L2 MRS 9B	7.92 ± 0.3	1.5 ± 0.2 ^a^	1.2 ± 0.1 ^b^
L3 MRS 5B	7.61 ± 0.3	1.3 ± 0.2 ^a^	1.5 ± 0.3 ^a^
L4 MRS 4B	7.74 ± 0.6	1.5 ± 0.2 ^a^	1.3 ± 0.2 ^a^

^1^ L: LAB. ^2^ Values indicate the average viable count of triplicate experiments with standard deviation in parentheses. Means in the same row with different letters were significantly different (*p* < 0.01).

**Table 4 foods-12-01182-t004:** Gene sequencing information from 16S rRNAs.

Isolate	Species	NCBI Accession No.
L4 M17 2A	*Enterococcus faecalis* strain 133170041-3	CP046108
L1 MRS 1B	*Lactobacillus helveticus* strain LH5	CP019581
L2 MRS 7B	*Lactobacillus delbrueckii* subsp. *indicus*	LC483566
L2 MRS 8B	*Lactobacillus delbrueckii* subsp. *sunkii* strain JCM 17838	CP018217
L2 MRS 9B	*Lactobacillus delbrueckii* subsp. *lactis* strain KCTC 3035	CP018156
L3 MRS 5B	*Lactobacillus delbrueckii* subsp. *sunkii* strain JCM 17838	CP018217
L4 MRS 4B	*Lactobacillus delbrueckii* subsp. *lactis strain KCTC 3035*	CP018156

*Lactobacillus delbrueckii* subsp. *sunkii* and *Lactobacillus delbrueckii* subsp. *lactis* were the most frequently isolated species. In addition, one isolate was identified as a member of the genus *Enterococcus*.

## Data Availability

Data is contained within the article.

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
