# Peer review of "Comparison of Activity of Commercial Protective Cultures and Thermophilic Lactic Acid Bacteria against Listeria monocytogenes: A New Perspective to Improve the Safety of Sardinian PDO Cheeses"

_foods, 2023, doi:10.3390/foods12061182_

Round 1

Reviewer 1 Report

Dear Editor and Authors,

The manuscript needs many modifications and corrections. 

1-Major comment

1-The title of the manuscript does not match the content of the manuscript, Is Listeria monocytogenes bacteria inhibited in Sardinian PDO cheeses?

2-Chapter Methods of work, the identification of lactic acid bacteria using several tests reported in page 3 line 135-137. These tests did not show any results in the results chapter. How does the reader know what the results are, how the isolates were chosen, and whether they are lactic acid bacteria or not? Vague and unclear results.

3- 16S rRNA test, this is an important test for which the authors did not mention any details such as what is the initiator used? What are the conditions of the PCR experiment? How were isolates identified? What is the extent of match with the reference isolates?

4-The use of Elliker agar to isolate Streptomyces thermophilus is not correct, because this medium grows all species of lactic acid bacteria, how was the differentiation between them? And the culture medium is not suitable for isolating Streptomyces thermophilus.

5-Tables 1a, 1b and 2 contain a statistical analysis, the authors did not mention anything about statistical analysis in the chapter on work methods.

2-Minor comment

1-The abstract of the manuscript needs to add some results.

2-The introduction needs to be supported by some scientific reference that shows the presence of Listeria monocytogenes bacteria in food, Such as 

Niamah, A. K. (2012). Detection of Listeria monocytogenes bacteria in four types of milk using PCR. Pakistan Journal of Nutrition, 11(12), 1158.‏

Yap, P. C., MatRahim, N. A., AbuBakar, S., & Lee, H. Y. (2021). Antilisterial potential of lactic acid bacteria in eliminating Listeria monocytogenes in host and ready-to-eat food application. Microbiology Research, 12(1), 234-257.

3-The modern nomenclature must be adopted when writing the names of bacteria belonging to lactic acid bacteria, such as Lactobacillus plantarum write Lactiplantibacillus plantarum.

4-Figure titles are always at the bottom of the figure, not at the top, see figure 1.

5-Conclusions Some results contain, it should be deleted and conclusions rewritten.

Author Response

Dear reviewer

According to your comments, we have updated the attached manuscript Comparison of activity of commercial protective cultures and thermophilic Lactic Acid Bacteria against Listeria monocytogenes: a new perspective to improve the safety of Sardinian PDO cheeses” (the title has been changed) and would like to resubmit it in the present form for the publication in “Foods”.

A language revision was made by one of the coauthor (Olivia McAuliffe, who is a native speaker).

This paper has not been published, accepted for publication or not under consideration at another journal. No other papers using the same data set have been published.

Reviewer 2 Report

Dear Author, I reviewed the manuscript (foods-2142158) entitled Comparison of commercial protective cultures and thermophilic Lactic Acid Bacteria activity against Listeria monocytogenes in Sardinian PDO cheeses. This manuscript presents relevant information about the antibacterial potential of lactic acid bacteria against L. monocytogenes. However, some sections of the presented data can be improved. For this reason, I consider that this manuscript needs minor changes. 

Additional comments.

Highlight the advantages of lactic acid bacteria against foodborne pathogens.

Check the paragraph extension in this manuscript.

Include an experimental design containing statistical factors and response variables in the statistical analyses applied to the findings of this research.

Try to include a statistical description in the required figures or tables. 

Compare the obtained findings with similar assays where lactic acid bacteria were used to inhibit pathogenic bacteria grown in similar food products. 

Include future trends to keep working with the obtained data. 

Try to conclude with a general statement of the most relevant part of this study.

Author Response

Dear reviewer,

According to your comments, we have updated the attached manuscript Comparison of activity of commercial protective cultures and thermophilic Lactic Acid Bacteria against Listeria monocytogenes: a new perspective to improve the safety of Sardinian PDO cheeses” (the title has been changed) and would like to resubmit it in the present form for the publication in “Foods”.

A language revision was made by one of the coauthor (Olivia McAuliffe, who is a native speaker).

This paper has not been published, accepted for publication or not under consideration at another journal. No other papers using the same data set have been published.

Reviewer 3 Report

This is an interesting study, in which autochthonous thermophilic LABs were isolated from raw sheep milk and were tested as protective cultures during the manufacturing of Sardinian PDO cheeses to improve their food safety, against Listeria monocytogenes. There are many cases of foodborne diseases due to this pathogen in EU and it is always important to publish new data concerning methods to reduce its presence in foods.

Some suggestions for improvement:

Line 18: The taxonomy has changed and Lactobacillus plantarum is called Lactiplantibacillus plantarum. Please correct throughout the manuscript

Line 39: “foodborn”, please change to “foodborne”

Line 80: “Consumers expect food to be safe, but they are aware of the possible adverse health effects of chemical additives in food”, this sentence seems irrelevant because there is no mention for chemical additives in the text that follows

Line 217: “lists” should change to “list”

Author Response

(The authors gave the same response as above.)

Round 2

Reviewer 1 Report

Dear  Editor

The authors have completed all necessary modifications to the manuscript, and it is now ready for publication.